# The effect of short-course point-of-care echocardiography training on the performance of **medical interns** in children

Esfandiar Nazari[1], Zahra Bahman Tajani[1], Saman Maroufizadeh[2],
Mohammad Ghorbani[1], Afagh Hassanzadeh Rad[1], Hamidreza Badeli[1]*

**1** Pediatric Diseases Research Center, Guilan University of Medical Sciences, Rasht, Iran, **2** Department of Biostatistics, School of Nursing and Midwifery, Guilan University of Medical Sciences, Rasht, Iran

* Badeli@gums.ac.ir

**Data Availability Statement:** All relevant data are within the paper and its supporting information files.

## Abstract

### Background

Point-of-care ultrasound (POCUS) can add complementary information to physical examination. Despite its development in several medical specialties, there is a lack of similar studies on children by medical interns and cardiologists. Therefore, investigators aimed to assess the effect of short-course training on the performance of medical interns in point-of-care echocardiography in children.

### Methods

This analytic cross-sectional study was conducted on 161 hospitalized children in 17 Shahrivar children's hospital, Iran, from January 2021 to May 2021. Seven interns (trainees) participated in a short course of point-of-care echocardiography to assess left ventricular ejection fraction (LVEF), inferior vena cava collapsibility index (IVCCI), and the presence of pericardial effusion (PEff). Each patient underwent point-of-care echocardiography by one of the trainees. Then, in less than one hour, the echocardiography was performed by a single cardiologist. Agreement between the cardiologist and trainees was examined using Cohen's kappa coefficient and Prevalence-Adjusted Bias-Adjusted Kappa (PABAK). For numerical variables, the agreement was examined using the concordance correlation coefficient (CCC) and intraclass correlation coefficient (ICC).

### Results

Results showed that the cardiologist and trainees detected LVEF >50, IVCCI >50%, and the absence of PEff in most of the participants. A good agreement in terms of ICC and CCC for LVEF (0.832 and 0.831, respectively) and a good agreement in terms of ICC and CCC for IVCCI (0.878 and 0.877, respectively) were noted. Using categorical scoring of LVEF and IVCCI showed 94.4% and 87.6% complete agreement, respectively. Furthermore, using categorical scoring of LVEF and IVCCI, Cohen's kappa coefficient was 0.542 (moderate) and 0.619 (substantial), respectively. The PABAK for LVEF and IVCCI were 0.886 (almost perfect) and 0.752 (substantial), respectively. For PEff, Cohen's kappa and PABAK were

**Funding:** The author(s) received no specific funding for this work.

**Competing interests:** The authors have declared that no competing interests exist.

**Abbreviations:** SD, Standard Deviation; CI, Confidence Interval; LVEF, Left Ventricular Ejection Fraction; IVCCI, Inferior Vena Cava Collapsibility Index; PABAK, Prevalence-Adjusted Bias-Adjusted Kappa; CCC, Concordance Correlation Coefficient; ICC, Intraclass Correlation Coefficient.

0.797 (moderate) and 0.988 (almost perfect), respectively, and the complete agreement was noted in 160 patients (99.4%).

## Conclusions

This study showed that a short teaching course could help medical interns to assess LVEF, IVCCI, and PEff in children. Therefore, it seems that adding this course to medical interns' curricula can be promising.

## Background

Point-of-care ultrasound (POCUS) is the application of ultrasound by non-radiologists that can add complementary, necessary, and rapid information to physical examination [1]. In recent decades, the use of ultrasound has been developed in diverse branches of medicine such as emergency medicine, gynecology, obstetrics, urology, nephrology, anesthesiology, etc [2].

The application of cardiovascular ultrasound at the bedside as a type of POCUS is highlighted as a complementary diagnostic tool in medical settings. Point-of-care echocardiography or focused cardiac ultrasound (FoCUS) helps clinicians with accurate diagnosis, early decision making, and appropriate treatment by assessing cardiac function and differentiating causes of shock, dyspnea, and chest pain [3]. Due to the lack of access to cardiologists in emergency settings, using FoCUS by non-radiologists can aid clinicians in having a rapid assessment and diagnosis [2]. It measures cardiac parameters such as left ventricular ejection fraction (LVEF), inferior vena cava collapsibility index (IVCCI), the presence of pericardial effusion (PEff), and tamponade.

Medical interns and residents in referral teaching hospitals are the first-line healthcare providers. Because of their persistent attendance at the patients' bedside, teaching technical skills for using FoCUS can lead to immediate decision making and accelerate the diagnosis and treatment process [4]. Previous investigations showed that using ultrasound in teaching anatomy, physiology, and pathophysiology courses enhanced basic science training in 1st- and 2nd-year students [5, 6].

It has facilitated learning the clinical skills in the 3rd- and 4th- year medical students [7, 8]. To now, limited studies on adults measured the agreement of measuring cardiac parameters by medical residents and clinicians [3, 9] and reported positive results. Also, recent studies have already assessed the efficacy of point-of-care transthoracic echocardiography training for medical students [10–12]. A similar previous study on children [13] by emergency medicine residents indicated promising results as well.

Despite the development of POCUS in several medical curricula [14, 15] focusing on its use by medical interns, there is a lack of similar studies on children. Therefore, we aimed to measure LVEF, IVCCI, and the presence of PEff through the application of FoCUS. As we could not assess children in critical situations and do multiple FoCUS exams on one patient (child) due to ethical considerations, this study investigated the effect of short-course training on the performance of medical interns in FoCUS for measuring the parameters mentioned above in a referral pediatric cardiology ward.

## Methods

### Patients and settings

This analytic cross-sectional study was conducted on 161 hospitalized children in 17 Shahrivar hospital, Iran, from January to May 2021. The inclusion criteria were patients' need for echocardiography in non-critical settings and patients' and/or parents' willingness to participate in

this study. Besides, we did not include patients with poor cooperation. Seven inexperienced in echocardiography interns (trainees) participated in performing FoCUS.

## Teaching course

Before enrollment, trainees attended a ten-hour course by two experienced clinicians, a pediatric cardiologist and a pediatric nephrologist (who had more than ten years of experience in POCUS). This course consisted of teaching knobology, modes, measuring IVC and LVEF diameters, and detecting the presence or absence of PEff. In addition to the ten-hour course, trainees observed five real-time point-of-care echocardiography by the cardiologist and practiced on five patients under the supervision of the cardiologist.

LVEF was measured by the M-mode method on the apex of the mitral valve in short and long axes views. The internal diameter of the left ventricle, intraventricular wall thickness, and left ventricular posterior wall thickness at the end of systole and diastole were measured. The percentage of LVEF was calculated by Teichholz [16] formula. Based on this formula, LVEF was divided into three subclasses of <30% (severe), 30–50% (moderate), and >50% (normal).

IVCCI was performed by the obtained images from the subcostal long-axis view. The IVC diameter was calculated in two centimeters from the junction of the right atrium and perpendicular to the long axis. Measuring the minimum diameter of inhalation (IVCmin) and the maximum diameter of exhalation (IVCmax) was mandatory to measure the IVCCI. IVCCI was defined based on the following formula: (IVCmax—IVCmin) / IVCmax ×100. It was classified as ≤50% (hypervolumic) and >50% (euvolumic or hypovolumic).

Regarding the presence or absence of PEff, the following method was used. PEff was diagnosed by the presence of an anechoic stripe in the pericardium around the heart and classified based on the presence or absence of effusion.

## Data gathering

In this study, each patient underwent FoCUS by one of the trainees. Then, in less than one hour, the echocardiography was performed by a single cardiologist. Trainees and the cardiologist used the same ultrasound device (Samsung EKO7) with a phase array probe and were unaware of the clinical diagnosis.

Data were gathered by a form including demographic characteristics (age and sex) and the concordance of obtained results between trainees and the cardiologist regarding the three cardiac parameters (LVEF, IVCCI, and PEff). The authors had access to information that could identify individual participants during or after data collection.

## Sample size

As this study aimed to assess the agreement between medical interns and the cardiologist, 161 patients were needed to be enrolled (23 patients for each intern).
$\alpha = 0.05$,
$z1-\alpha/2 = 1.96$
ρ Intraclass Correlation Coefficient (the expected reliability):0.8
m (numbers of trainees): 7
W (Weighted Kappa Coefficient): 0.3

## Ethical considerations

Ethics approval was obtained from the Regional Ethics Committee, Guilan University of Medical Sciences. (No: IR.GUMS.REC.1399.477, Date: 1.6.2021). The informed written consent letter was obtained from the parents or guardians before enrollment.

## Statistical analysis

Data were reported by number, percent, mean, and standard deviation. LVEF and IVCCI were analyzed numerically and categorical, and PEff was indicated only as a nominal variable. Inter-rater agreement between cardiologists and trainees was calculated using Cohen's kappa coefficient for qualitative variables. The Prevalence-Adjusted Bias-Adjusted Kappa (PABAK) was also calculated to address the influence of prevalence and bias on Cohen's kappa. For numerical variables, inter-rater agreement was examined using concordance correlation coefficient (CCC), intraclass correlation coefficient (ICC), and Bland-Altman Plot. ICC as a reliability index reflects both degrees of correlation and agreement. The CCC measures the degree to which pairs of observations fall on the 45° line through the origin. The scatter plot points will line up near the 45° line through the origin if the measurements agree closely. The strength of agreement for the kappa value and PABAK can be interpreted as follows: <0 = poor; 0–0.20 = slight; 0.21–0.40 = fair; 0.41–0.60 = moderate; 0.61–0.80 = substantial; 0.81–1.00 = almost perfect. For ICC and CCC, values of <0.5, 0.5–0.75, 0.75–0.9, and >0.9 were considered as poor, moderate, good, and excellent agreement, respectively [17]. Data analysis was performed using MedCalc for Windows, version 18.9.1 (MedCalc Software, Ostend, Belgium), and graphs were depicted using GraphPad Prism, Version 8.0.1 (GraphPad Prism Software Inc., San Diego, CA, USA).

## Results

Of the 161 children, 86 (53.4%) were boys, and the mean age was 5.86 ± 3.83 years. Results showed that the cardiologist and trainees indicated LVEF >50% (92.6% versus 94.4%, respectively), IVCCI >50% (80.2% versus 87.9%, respectively), and the absence of PEff (98.1% versus 98.8%, respectively) in the majority of participants (Table 1).

## Left Ventricular Ejection Fraction (LVEF)

The CCC was estimated to be 0.831 (good agreement, 95% CI: 0.776 to 0.873), and the scatter plot in Fig 1A shows that the data points fell on or near the line of equality. The ICC was 0.832 (good agreement, 95% CI: 0.777 to 0.874). we also assessed the agreement between the

**Table 1. The descriptive statistics for LVEF, IVCCI, and PE among children according to the cardiologist and trainees' assessments.**

|  | Cardiologist | Trainees |
|---|---|---|
| **LVEF**, n (%) |  |  |
| <30 | 0 | 0 |
| 30–50 | 12 (7.4) | 9 (5.6) |
| >50 | 149 (92.6) | 152 (94.4) |
| Mean ± SD | 64.98 ± 8.85 | 64.90 ± 8.63 |
| **IVCCI**, n (%) |  |  |
| <50 | 32 (19.8) | 34 (21.1) |
| >50 | 129 (80.2) | 127 (87.9) |
| Mean ± SD | 59.24 ± 11.42 | 59.43 ± 11.65 |
| **Pericardial Effusion**, n (%) |  |  |
| Negative | 159 (98.8) | 158 (98.1) |
| Positive | 2 (1.2) | 3 (1.9) |

SD: Standard Deviation; LVEF: Left Ventricular Ejection Fraction; IVCCI: Inferior Vena Cava Collapsibility Index.

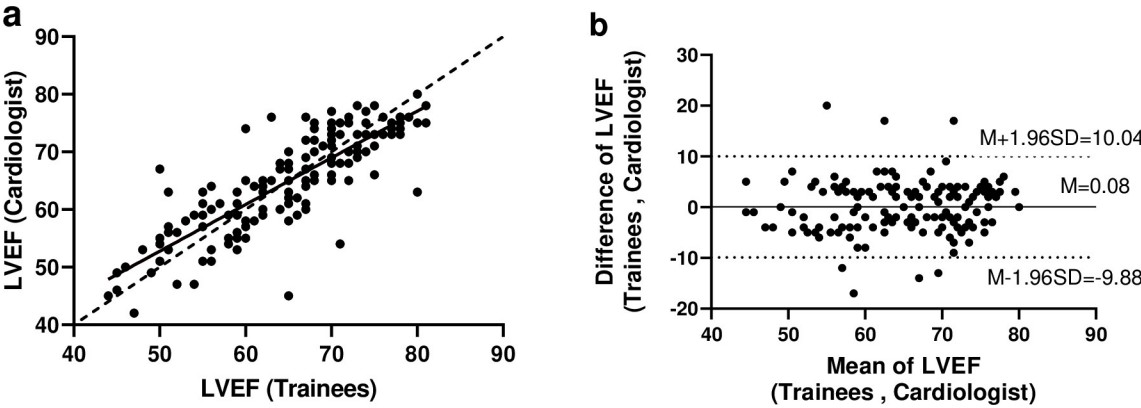

**Fig 1.** (a) Scatter plot of cardiologist versus trainees on LVEF, (b) Bland and Altman plot for assessing agreement between trainees and cardiologist on LVEF. LVEF: Left Ventricular Ejection Fraction. Note. (a) Solid line: best-fit line; Dashed line: 45° line through the origin. (b) M: Mean; SD: Standard Deviation.

cardiologist and trainees through the visual inspection of the Bland-Altman plot (Fig 1B). This figure shows that nearly all the points (95.6%) lie between the lower and upper limits of agreement. No bias or trend in the differences can be observed, indicating good agreement between the cardiologist and trainees. Furthermore, using categorical scoring of LVEF, the complete agreement was 94.4% (152/161), and Cohen's kappa coefficient was 0.542 (95% CI: 0.278 to 0.807), which was considered to be moderate agreement (see Table 2). The PABAK that addressed the influence of low prevalence on Cohen's kappa was 0.886, which was considered to be almost perfect.

## Inferior Vena Cava Collapsibility Index (IVCCI)

The CCC was 0.877 (good agreement, 95% CI: 0.836 to 0.908). Fig 2A also shows the points lined up close to the 45° line through the origin. The ICC was 0.878 (good agreement, 95% CI: 0.837 to 0.909). The Bland-Altman plot (Fig 2B) shows that all the points except seven (95.6%) were within the lower and upper limits of agreement; No obvious trend/bias was found in the scattering of points. Furthermore, using categorical scoring of IVCCI, the complete agreement was 87.6% (141/161), and Cohen's kappa coefficient was 0.619 (95% CI: 0.467 to 0.771), which was considered to be moderate agreement (see Table 3). The PABAK was 0.752, which was considered to be substantial.

**Table 2. Agreement between cardiologist and trainees in assessing left ventricular ejection fraction among children.**

|  |  | Cardiologist |  |
| --- | --- | --- | --- |
|  |  | LVEF 30–50 | LVEF 51–100 |
| Trainees | LVEF 30–50 | 6 | 3 |
|  | LVEF 51–100 | 6 | 146 |
| Cohen's kappa coefficient (95% CI) |  | 0.542 (0.278–0.807) |  |
| PABAK |  | 0.886 |  |
| Complete agreement, n (%) |  | 152 (94.4%) |  |

LVEF: Left Ventricular Ejection Fraction; CI: Confidence Interval; PABAK: Prevalence-Adjusted Bias-Adjusted Kappa.

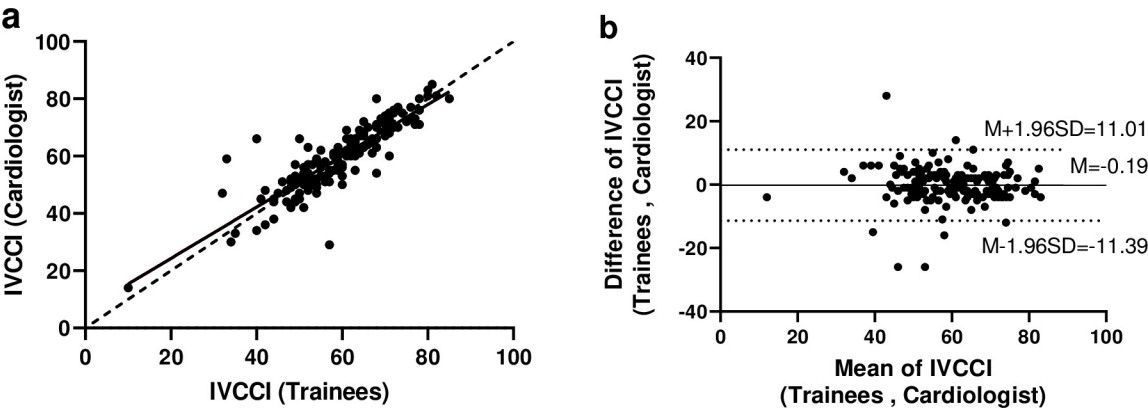

**Fig 2.** (a) Scatter plot of cardiologist versus trainees on IVCCI, (b) Bland and Altman plot for assessing agreement between trainees and cardiologist on IVCCI. IVCCI: Inferior Vena Cava Collapsibility Index. Note. (a) Solid line: best-fit line; Dashed line: 45° line through the origin. (b) M: Mean; SD: Standard Deviation.

### Pericardial Effusion (PE)

The complete agreement was noted in 160 patients (99.4%). The Cohen kappa coefficient was 0.797 (95% CI: 0.409–1.000), indicating substantial agreement. The PABAK was 0.988 (95% CI: 0.963–1.000), considered almost perfect (Table 4).

## Discussion

FoCUS can add complementary information to routine physical examinations. It is a known concept for emergency medicine specialists [18–20]. Pervious limited investigations assessed point-of-care echocardiography training in emergency and internal medicine, and pediatric residents reported promising results [3, 9]. Besides, this study showed that a limited teaching course could help medical interns to perform FoCUS to assess LVEF, IVCCI, and PEff in children.

In the current study, there was an excellent agreement (94.4%) between the cardiologist and interns regarding LVEF. Bustam et al. performed FOCUS by the emergency medicine residents on 100 adult patients and mentioned a 92.9% agreement between residents and the cardiologist for LVEF [3]. Randazzo et al. reported 86.1% as the agreement of FOCUS performed by eight non-cardiologist clinicians [9]. Considering the agreement levels, it seems that our medical interns' ability was the same as individuals with higher educational levels in the

**Table 3. Agreement between cardiologist and trainees in assessing inferior vena cava collapsibility index among children.**

| | | Cardiologist | |
|---|---|---|---|
| | | **IVCCI 1–50** | **IVCCI 51–100** |
| Trainees | IVCCI 1–50 | 23 | 11 |
| | IVCCI 51–100 | 9 | 118 |
| Cohen's kappa coefficient (95% CI) | | 0.619 (0.467–0.771) | |
| PABAK | | 0.752 | |
| Complete agreement, n (%) | | 141 (87.6%) | |

IVCCI: Inferior Vena Cava Collapsibility Index; CI: Confidence Interval; PABAK: Prevalence-Adjusted Bias-Adjusted Kappa.

**Table 4. Agreement between cardiologist and trainees in assessing pericardial effusion among children.**

| | | Cardiologist | |
| --- | --- | --- | --- |
| | | **PE -** | **PE +** |
| Trainees | PE - | 158 | 0 |
| | PE + | 1 | 2 |
| Cohen's kappa coefficient (95% CI) | | 0.797 (0.409–1.000) | |
| PABAK | | 0.988 | |
| Complete agreement, n (%) | | 160 (99.4%) | |

PE: Pericardial Effusion; CI: Confidence Interval; PABAK: Prevalence-Adjusted Bias-Adjusted Kappa

previous investigations [3, 9]. Hüppe et al. performed FOCUS on 250 patients by 25 of last year's medical interns (ten patients by each medical intern). Their results showed that the mean agreement of assessing LVEF in the first patient was 60%, reaching 91.3% in the tenth patient [21]. Their results emphasized the importance of exercise in improving their skills.

Regarding IVCCI, the results showed a good agreement (87.6%) between the trainees and the cardiologist. Bustam et al. assessed the IVCCI and mentioned a 64.2% agreement (moderate level) between the emergency medicine residents and the cardiologist [3]. As IVC diameter can be influenced by respiratory rate and inhalation volume; it may induce this difference in concordance level between the cardiologist and trainees.

The agreement regarding the presence of PEff between the cardiologist and trainees was 99.4%. Consistent with our results, Bustam et al. mentioned 98% agreement [3]. Although there were limited cases of PEff in our participants, due to the complete agreement, we can conclude that the trainees and the cardiologist had practical knowledge to rule out the absence of PEff.

## Limitations

Although performing multiple FoCUS exams on one patient could ideally assess the agreement between trainees and the cardiologist, we evaluated each patient by one intern and cardiologist due to ethical considerations. Based on our study design and exclusion criteria, we enrolled 161 patients by consecutive sampling. Therefore, we could not allocate patients with diverse types of LVEF, IVCCI, and PEff. Hence, it would be better to design further investigations focusing on pediatric patients with definite pathologies.

## Conclusions

This study showed that a short teaching course could help medical interns to assess LVEF, IVCCI, and PEff in children. Therefore, it seems that adding this course to medical interns' curricula can be promising. Further studies are recommended to determine how integrating medical interns in the imaging process can change clinical treatment and improve efficiency or time-to-treatment.

## Supporting information

**S1 Checklist.** *PLOS ONE* **clinical studies checklist.**
(DOCX)

**S2 Checklist. STROBE statement—checklist of items that should be included in reports of observational studies.**
(DOCX)

**S1 Data.**
(XLSX)

## Acknowledgments

This study was approved as a proposal by the Vice-Chancellor of Guilan University of Medical Sciences. It was the thesis of the second author (Dr. Zahra Bahman Tajani).

## Author Contributions

**Conceptualization:** Esfandiar Nazari, Zahra Bahman Tajani, Saman Maroufizadeh, Mohammad Ghorbani, Afagh Hassanzadeh Rad, Hamidreza Badeli.

**Data curation:** Zahra Bahman Tajani, Saman Maroufizadeh, Hamidreza Badeli.

**Investigation:** Zahra Bahman Tajani, Mohammad Ghorbani, Afagh Hassanzadeh Rad, Hamidreza Badeli.

**Methodology:** Esfandiar Nazari, Saman Maroufizadeh, Mohammad Ghorbani, Afagh Hassanzadeh Rad, Hamidreza Badeli.

**Project administration:** Hamidreza Badeli.

**Supervision:** Saman Maroufizadeh, Hamidreza Badeli.

**Validation:** Esfandiar Nazari, Hamidreza Badeli.

**Visualization:** Zahra Bahman Tajani, Hamidreza Badeli.

**Writing – original draft:** Esfandiar Nazari, Zahra Bahman Tajani, Saman Maroufizadeh, Mohammad Ghorbani, Afagh Hassanzadeh Rad, Hamidreza Badeli.

**Writing – review & editing:** Esfandiar Nazari, Zahra Bahman Tajani, Saman Maroufizadeh, Mohammad Ghorbani, Afagh Hassanzadeh Rad, Hamidreza Badeli.

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
