## [Decision Letter · Decision Letter 0]

6 Jun 2022

PONE-D-22-05938The effect of short course training on the performance of medical students in point-of-care echocardiography: A cross-sectional study.PLOS ONE

Dear Dr. Badeli ,

Thank you for submitting your manuscript to PLOS ONE. After careful consideration, we feel that it has merit but does not fully meet PLOS ONE’s publication criteria as it currently stands. Therefore, we invite you to submit a revised version of the manuscript that addresses the points raised during the review process.

ACADEMIC EDITOR:

Dear Authors

First of all, I apologize for the delay.

I would like you to try answering all the question address by the Reviewer in order to re-evaluate your manuscript.

We look forward to receiving your revised manuscript.

Kind regards,

Manuela Cabiati, Ph.D.

Academic Editor

PLOS ONE

Journal Requirements:

3. You indicated that you had ethical approval for your study. In your Methods section, please ensure you have also stated whether you obtained consent from parents or guardians of the minors included in the study or whether the research ethics committee or IRB specifically waived the need for their consent.

Reviewers' comments:

Reviewer's Responses to Questions

**Comments to the Author**

1. Is the manuscript technically sound, and do the data support the conclusions?

Reviewer #1: Yes

Reviewer #2: Partly

2. Has the statistical analysis been performed appropriately and rigorously? 

Reviewer #1: Yes

Reviewer #2: I Don't Know

3. Have the authors made all data underlying the findings in their manuscript fully available?

Reviewer #1: Yes

Reviewer #2: Yes

4. Is the manuscript presented in an intelligible fashion and written in standard English?

Reviewer #1: No

Reviewer #2: No

5. Review Comments to the Author

Reviewer #1: I am very grateful for the opportunity to review this manuscript.

The study is about the agreement between the point of care ultrasound findings between trained medical interns and a professional cardiologist. The study is interesting and provides good information about the topic. I have a few points to address.

Despite the study was conducted in a pediatric population, this was not mentioned either in the title or in the conclusion of the manuscript. The title has to clearly state that this is a pediatric study, to attract the correct audience. In addition, the conclusion should be based only on the study results, so it has to mention the pediatric population in any drawn conclusion.

All over the study, the authors exchange between medical students and medical interns. It is better to stick to the description of medical interns as they are obviously different from medical students.

The overall language of the manuscript needs a thorough revision, as some sentences are not clear and difficult to understand and other have grammatical errors. Some examples are mentioned below:

Abstract

The first sentence of the abstract is not very clear and the second one is very lengthy and difficult to understand.

The second sentence of the abstract results better to be rephrased for example to start with “A good agreement in terms of ... and a good agreement of ... were noted”

Keywords: could be more specific and contain the words POCUS and FoCUS

Background

- “… to obtain results from the physical examination”, What does that mean? Do you mean to add data to the physical examination?

- “According to the lack of access”. I think you mean due to lack of access.

- “…in referral educational and care centers”, needs further explanation

- “Regarding their persistent …”, you can change to “because of …”

- “To now, limited studies have been performed to assess the agreement of measuring cardiac parameters by medical residents and clinicians on adults”, needs rephrasing

- “The development of POCUS in several medical curricula focusing on its use by medical students, the rapid development of incorporating it in medicine, and the lack of similar study in pediatrics assessing the effect of limited training on medical interns”. The sentence is lengthy and unclear.

Methods

- “short and long axes” , add “ views

- “two centimeters of the junction from the right atrium”, should be “from the junction”

- Please add more details about the device model and type of the probe used.

- “As this study aimed to assess the agreement …”. Please add the agreement between whom

Results

- “indicated LVEF >50%”, change to” indicated that LVEF was >50%”

- “most patients”, should be “most of the patients”

- “was considered almost perfect”, use “to be”

Discussion

- “and some medical centers have already used it routinely”, The sentence needs a reference

- “Based on the importance of this issue, this study showed that a limited teaching course could help medical interns to perform FoCUS for assessing LVEF”. The sentence structure needs to be changed

- “Results showed that the ability of medical interns was the same with residents and clinicians with higher educational levels”. Which results are you referring to here?

- “had practical knowledge”, should be “had practical knowledge”

Reviewer #2: Nazari et al have performed a cross-sectional study conducted on 161 hospitalized children in 17 Shahrivar children's hospital, Iran, from January 2021 to May 2021. Seven interns (trainees) participated in a short course of point-of-care echocardiography for assessing left ventricular ejection fraction (LVEF), inferior vena cava collapsibility index (IVCCI), and the presence of pericardial effusion (PEff). Each echocardiographic analysis was performed by one of the trainees and a single cardiologist.Agreement between the cardiologist and trainees was examined using Cohen's kappa coefficient and Prevalence-Adjusted Bias-Adjusted Kappa (PABAK). Results showed that the cardiologist and trainees detected LVEF >50, IVCCI >50%, and the absence of PEff in most of the participants. The authors conclude that short teaching course could help medical interns to assess LVEF, IVCCI, and PEff in an accurate manner.

Major issues

1) Recent studies has already assessed the efficacy of a point-of-care transthoracic echocardiography training for medical students (please see J Cardiothorac Vasc Anesth. 2021 Mar;35(3):826-833; Pilot Feasibility Stud. 2021 Sep 14;7(1):175; Postgrad Med J. 2021 Jan;97(1143):10-15. ). The authors do not mention these studies. Therefore, they should evaluate the novelty of the present study in the light of the previous ones.

2) The manuscript requires a comprehensive editing in order to better discuss the results. The reader sometimes has the impression of reading a preliminary report without adequate elaboration.

3) In the current study, the authors claims that there was an excellent agreement (94.4%) between the cardiologist and trainees regarding LVEF.The percentage of LVEF was calculated by Teichholz formula. However, the authors should show data regarding LV end-diastolic and end-systolic internal diameters, and heart rate. Moreover, data on intraventricular wall thickness, and left ventricular posterior wall thickness at the end of systole and diastole should be provided. What about the rhythm? Please add more information.

6. PLOS authors have the option to publish the peer review history of their article (what does this mean?). If published, this will include your full peer review and any attached files.

Reviewer #1: **Yes: **Mohammed Abdellatif

Reviewer #2: No

---

## [Author Response · Author response to Decision Letter 0]

20 Jul 2022

Dear editor in chief, 

We appreciate your great comments, we revised our article based on these comments. 

We appreciate your comment. We modified and added the following statement regarding the consent letter. The informed written consent letter was obtained from the parents or guardians before enrollment. 

3. You indicated that you had ethical approval for your study. In your Methods section, please ensure you have also stated whether you obtained consent from parents or guardians of the minors included in the study or whether the research ethics committee or IRB specifically waived the need for their consent.

We appreciate your comment. We modified and added the following statement regarding the consent letter. The informed written consent letter was obtained from the parents or guardians before enrollment. 

We appreciate your comment. We deleted it. 

We modified it. 

Reviewers' comments:

Reviewer #1: 

 Despite the study was conducted in a pediatric population, this was not mentioned either in the title or in the conclusion of the manuscript. The title has to clearly state that this is a pediatric study, to attract the correct audience. In addition, the conclusion should be based only on the study results, so it has to mention the pediatric population in any drawn conclusion.

We appreciate your comment. We modified and added the following statement. 

Title: The effect of short-course point-of-care echocardiography training on the performance of medical interns in children.

Conclusion: This study showed that a short teaching course could help medical interns to assess LVEF, IVCCI, and PEff in children. Therefore, it seems that adding this course to medical inerns' curricula can be promising.

All over the study, the authors exchange between medical students and medical interns. It is better to stick to the description of medical interns as they are obviously different from medical students.

We appreciate your comment. We modified it and mentioned medical interns in all sections related to our study. 

The overall language of the manuscript needs a thorough revision, as some sentences are not clear and difficult to understand and other have grammatical errors. Some examples are mentioned below:

Abstract

The first sentence of the abstract is not very clear and the second one is very lengthy and difficult to understand.

We appreciate your comment. We modified it. Point-of-care ultrasound (POCUS) can add complementary information to physical examination. Despite its development in several medical specialties, there is a lack of similar studies on children by medical interns and cardiologists. Therefore, investigators aimed to assess the effect of short-course training on the performance of medical interns in point-of-care echocardiography in children.

The second sentence of the abstract results better to be rephrased for example to start with “A good agreement in terms of ... and a good agreement of ... were noted”

We appreciate your comment. We modified it as below.

A good agreement in terms of ICC and CCC for LVEF (0.832 and 0.831, respectively) and a good agreement in terms of ICC and CCC for IVCCI (0.878 and 0.877, respectively) were noted.

Keywords: could be more specific and contain the words POCUS and FoCUS

We appreciate your comment. We searched MESH and unfortunately we could not find POCUS or FoCUS. Therefore, we added Ultrasonography and Diagnostic Imaging.

Background

- “… to obtain results from the physical examination”, What does that mean? Do you mean to add data to the physical examination?

We appreciate your comment. We modified it. 

Point-of-care ultrasound (POCUS) is the application of ultrasound by non-radiologists that can add complementary, necessary, and rapid information to physical examination

- “According to the lack of access”. I think you mean due to lack of access.

We appreciate your comment. We modified it. 

- “…in referral educational and care centers”, needs further explanation

We appreciate your comment. We modified it as referral teaching hospitals

- “Regarding their persistent …”, you can change to “because of …”

We appreciate your comment. We modified it. 

- “To now, limited studies have been performed to assess the agreement of measuring cardiac parameters by medical residents and clinicians on adults”, needs rephrasing

We appreciate your comment. We modified it as below.

To now, limited studies on adults measured the agreement of measuring cardiac parameters by medical residents and clinicians

- “The development of POCUS in several medical curricula focusing on its use by medical students, the rapid development of incorporating it in medicine, and the lack of similar study in pediatrics assessing the effect of limited training on medical interns”. The sentence is lengthy and unclear.

We appreciate your comment. We modified it as below.

Despite the development of POCUS in several medical curricula 11-12 focusing on its use by medical interns, there is a lack of similar studies on children.

Methods

- “short and long axes” , add “ views

We appreciate your comment. We modified it.

- “two centimeters of the junction from the right atrium”, should be “from the junction”

We appreciate your comment. We modified it as below.

From the junction of the right atrium

- Please add more details about the device model and type of the probe used.

We appreciate your comment. We modified it as below.

Ultrasound device (Samsung EKO7) with phase array probe

- “As this study aimed to assess the agreement …”. Please add the agreement between whom

We appreciate your comment. We modified it as below.

As this study aimed to assess the agreement between medical interns and the cardiologist.

Results

- “indicated LVEF >50%”, change to” indicated that LVEF was >50%”

We appreciate your comment. We modified it.

- “most patients”, should be “most of the patients”

We appreciate your comment. We modified it.

- “was considered almost perfect”, use “to be”

We appreciate your comment. We modified it.

Discussion

- “and some medical centers have already used it routinely”, the sentence needs a reference

We appreciate your comment. We added following statement and references.

FoCUS can add complementary information to routine physical examination. It is a known concept for emergency medicine specialists.

Farsi D, Hajsadeghi S, Hajighanbari MJ, Mofidi M, Hafezimoghadam P, Rezai M, Mahshidfar B, Abiri S, Abbasi S. Focused cardiac ultrasound (FOCUS) by emergency medicine residents in patients with suspected cardiovascular diseases. J Ultrasound. 2017 May 2;20(2):133-138. doi: 10.1007/s40477-017-0246-5. PMID: 28593003; PMCID: PMC5440337.

Biais M, Carrié C, Delaunay F, Morel N, Revel P, Janvier G. Evaluation of a new pocket echoscopic device for focused cardiac ultrasonography in an emergency setting. Crit Care. 2012 May 14;16(3):R82. doi: 10.1186/cc11340. PMID: 22583539; PMCID: PMC3580625.

Albaroudi B, Haddad M, Albaroudi O, Abdel-Rahman ME, Jarman R, Harris T. Assessing left ventricular systolic function by emergency physician using point of care echocardiography compared to expert: systematic review and meta-analysis. Eur J Emerg Med. 2022 Feb 1;29(1):18-32. doi: 10.1097/MEJ.0000000000000866. PMID: 34406134; PMCID: PMC8691376.

- “Based on the importance of this issue, this study showed that a limited teaching course could help medical interns to perform FoCUS for assessing LVEF”. The sentence structure needs to be changed

We appreciate your comment. We modified it as below.

Besides, this study showed that a limited teaching course could help medical interns to perform FoCUS for assessing LVEF, IVCCI, and PEff in children.

- “Results showed that the ability of medical interns was the same with residents and clinicians with higher educational levels”. Which results are you referring to here?

We appreciate your comment. We modified it as below.

Considering the agreement levels, it seems that the ability of our medical interns was the same with individuals with higher educational levels in the previous investigations 3,9

Reviewer #2: 

Major issues

1) Recent studies has already assessed the efficacy of a point-of-care transthoracic echocardiography training for medical students (please see J Cardiothorac Vasc Anesth. 2021 Mar;35(3):826-833; Pilot Feasibility Stud. 2021 Sep 14;7(1):175; Postgrad Med J. 2021 Jan;97(1143):10-15. ). The authors do not mention these studies. Therefore, they should evaluate the novelty of the present study in the light of the previous ones.

We appreciate your comment. We modified it and added the above mentioned studies. 

2) The manuscript requires a comprehensive editing in order to better discuss the results. The reader sometimes has the impression of reading a preliminary report without adequate elaboration.

We appreciate your comment. But as you know, this study aimed to assess the concordance between the cardiologist and medical interns regarding the three mentioned parameters and most of our results primarily reported as descriptive statistics. Although we tried to discuss our results clinically and changed some sections due to your valuable comments, unfortunately the nature of our results did not let us elaborate more in some sections.

3) In the current study, the authors claims that there was an excellent agreement (94.4%) between the cardiologist and trainees regarding LVEF. The percentage of LVEF was calculated by Teichholz formula. However, the authors should show data regarding LV end-diastolic and end-systolic internal diameters, and heart rate. Moreover, data on intraventricular wall thickness, and left ventricular posterior wall thickness at the end of systole and diastole should be provided. What about the rhythm? Please add more information.

We appreciate your great comment, but we have some ethical limitations. As you know, this preliminary study was performed by medical interns on children and indicated promising results. Despite thorough assessment of the cardiologist on each patient considering all items mentioned above, our medical interns only indicated LV end-diastolic and end-systolic diameters and our device automatically calculated LVEF by Teichholz formula. Therefore, we do not have an access to the detailed data. Besides, maybe it would be unethical to perform such a comprehensive assessment by a medical intern on a child. Certainly, a similar study including your items by individuals with higher level of education can be very helpful. We added your valuable comment as our recommendation for further studies.

---

## [Decision Letter · Decision Letter 1]

16 Aug 2022

PONE-D-22-05938R1The effect of short-course point-of-care echocardiography training on the performance of medical interns in childrenPLOS ONE Please submit your revised manuscript by Sep 30 2022 11:59PM. If you will need more time than this to complete your revisions, please reply to this message or contact the journal office at plosone@plos.org. Please include the following items when submitting your revised manuscript:A rebuttal letter that responds to each point raised by the academic editor and reviewer(s). You should upload this letter as a separate file labeled 'Response to Reviewers'.A marked-up copy of your manuscript that highlights changes made to the original version. You should upload this as a separate file labeled 'Revised Manuscript with Track Changes'.An unmarked version of your revised paper without tracked changes. You should upload this as a separate file labeled 'Manuscript'.

We look forward to receiving your revised manuscript.

Kind regards,

Manuela Cabiati, Ph.D.

Academic Editor

PLOS ONE

Additional Editor Comments:

Dear author

As you noted one of the Reviewer retains your article not suitable for publication in PlosOne since him/her comments remain unsolved or not well addressed. If you answer to hos/her issues more accurately I will reconsider my position. Best regards

Reviewers' comments:

Reviewer's Responses to Questions

**Comments to the Author**

1. If the authors have adequately addressed your comments raised in a previous round of review and you feel that this manuscript is now acceptable for publication, you may indicate that here to bypass the “Comments to the Author” section, enter your conflict of interest statement in the “Confidential to Editor” section, and submit your "Accept" recommendation.

Reviewer #1: All comments have been addressed

Reviewer #2: (No Response)

2. Is the manuscript technically sound, and do the data support the conclusions?

Reviewer #1: Yes

Reviewer #2: No

3. Has the statistical analysis been performed appropriately and rigorously? 

Reviewer #1: Yes

Reviewer #2: No

4. Have the authors made all data underlying the findings in their manuscript fully available?

Reviewer #1: Yes

Reviewer #2: No

5. Is the manuscript presented in an intelligible fashion and written in standard English?

Reviewer #1: Yes

Reviewer #2: No

6. Review Comments to the Author

Reviewer #1: Many thanks to the authors for addressing the comments

I think the manuscript is ready for publication.

Good luck!

Reviewer #2: Unfortuntaley, my questions remain unsolved. This reviewer considers these issues relevant to support the conclusions.

Regarding data, you should have collected the data at the time of the analysis. I'm really surprised that you didn't ask the question at the time of the patient evaluation.

7. PLOS authors have the option to publish the peer review history of their article (what does this mean?). If published, this will include your full peer review and any attached files.

Reviewer #1: **Yes: **Mohammed Abdellatif

Reviewer #2: No

---

## [Author Response · Author response to Decision Letter 1]

29 Oct 2022

Dear editor in chief, 

We appreciate our first reviewer regarding his acceptance on our revised manuscript. Also, we acknowledge our second reviewer for his great concern. In response to his/her statement, we rechecked her/his previous comments and you can find our revisions as below. 

Review Comments to the Author

Reviewer #1: Many thanks to the authors for addressing the comments

I think the manuscript is ready for publication.

Good luck!

We appreciate it. 

Reviewer #2: Unfortuntaley, my questions remain unsolved. This reviewer considers these issues relevant to support the conclusions.

Regarding data, you should have collected the data at the time of the analysis. I'm really surprised that you didn't ask the question at the time of the patient evaluation.

We appreciate your comment. We put your previous comments and our responses below. 

Reviewer #2: 

Major issues

1) Recent studies has already assessed the efficacy of a point-of-care transthoracic echocardiography training for medical students (please see J Cardiothorac Vasc Anesth. 2021 Mar;35(3):826-833; Pilot Feasibility Stud. 2021 Sep 14;7(1):175; Postgrad Med J. 2021 Jan;97(1143):10-15. ). The authors do not mention these studies. Therefore, they should evaluate the novelty of the present study in the light of the previous ones.

We appreciate your comment. We modified it and added the above mentioned studies in our references. 

The main novelty of our manuscript can be noted as follows: Despite the development of POCUS in several medical curricula focusing on its use by medical interns, there is a lack of similar studies on children. Therefore, we aimed to measure LVEF, IVCCI, and the presence of PEff through the application of FoCUS. As we could not assess children in critical situations and do multiple FoCUS exams on one patient (child) due to ethical considerations, this study investigated the effect of short-course training on the performance of medical interns in FoCUS for measuring the parameters mentioned above in a referral pediatric cardiology ward.

2) The manuscript requires a comprehensive editing in order to better discuss the results. The reader sometimes has the impression of reading a preliminary report without adequate elaboration.

We appreciate your comment. But as you know, this study aimed to assess the concordance between the cardiologist and medical interns regarding the three mentioned parameters and most of our results primarily reported as descriptive statistics. We tried to discuss our results clinically and changed some sections due to your valuable comments.

3) In the current study, the authors claims that there was an excellent agreement (94.4%) between the cardiologist and trainees regarding LVEF. The percentage of LVEF was calculated by Teichholz formula. However, the authors should show data regarding LV end-diastolic and end-systolic internal diameters, and heart rate. Moreover, data on intraventricular wall thickness, and left ventricular posterior wall thickness at the end of systole and diastole should be provided. What about the rhythm? Please add more information.

We appreciate your great comment, but we have some ethical limitations. As you know, this preliminary study was performed by medical interns on children and indicated promising results. Despite thorough assessment of the cardiologist on each patient considering all items mentioned above, our medical interns only indicated LV end-diastolic and end-systolic diameters and our device automatically calculated LVEF by Teichholz formula. Therefore, we do not have an access to the detailed data. Besides, maybe it would be unethical to perform such a comprehensive assessment by a medical intern on a child. Certainly, a similar study including your items by individuals with higher level of education can be very helpful. We added your valuable comment as our recommendation for further studies.

---

## [Editor Report · Decision Letter 2]

11 Nov 2022

The effect of short-course point-of-care echocardiography training on the performance of medical interns in children.

PONE-D-22-05938R2

Dear Dr. Badeli,

We’re pleased to inform you that your manuscript has been judged scientifically suitable for publication and will be formally accepted for publication once it meets all outstanding technical requirements.

Kind regards,

Manuela Cabiati, Ph.D.

Academic Editor

PLOS ONE
---

## [Editor Report · Acceptance letter]

5 Dec 2022

PONE-D-22-05938R2 

The effect of short-course point-of-care echocardiography training on the performance of <bold>medical interns<bold/> in children. 

Dear Dr. Badeli:

I'm pleased to inform you that your manuscript has been deemed suitable for publication in PLOS ONE. Congratulations! Your manuscript is now with our production department. 

Kind regards, 

on behalf of

Dr. Manuela Cabiati 

Academic Editor

PLOS ONE